# Characterization of Trehalose-6-Phosphate Synthase and Trehalose-6-Phosphate Phosphatase Genes of Tomato (*Solanum lycopersicum* L.) and Analysis of Their Differential Expression in Response to Temperature

**DOI:** 10.3390/ijms231911436

**Published:** 2022-09-28

**Authors:** Mohanna Mollavali, Frederik Börnke

**Affiliations:** 1Plant Metabolism Group, Leibniz-Institute of Vegetable and Ornamental Crops (IGZ), Theodor-Echtermeyer Weg 1, 14979 Großbeeren, Germany; 2Institute of Biochemistry and Biology, Campus Golm Haus 20, University of Potsdam, Karl-Liebknecht-Str. 24/25, 14476 Potsdam, Germany

**Keywords:** trehalose metabolism, heat stress, *Solanum lycopersicum*, yeast complementation

## Abstract

In plants, the trehalose biosynthetic pathway plays key roles in the regulation of carbon allocation and stress adaptation. Engineering of the pathway holds great promise to increase the stress resilience of crop plants. The synthesis of trehalose proceeds by a two-step pathway in which a trehalose-phosphate synthase (TPS) uses UDP-glucose and glucose-6-phosphate to produce trehalose-6 phosphate (T6P) that is subsequently dephosphorylated by trehalose-6 phosphate phosphatase (TPP). While plants usually do not accumulate high amounts of trehalose, their genome encodes large families of putative trehalose biosynthesis genes, with many members lacking obvious enzymatic activity. Thus, the function of putative trehalose biosynthetic proteins in plants is only vaguely understood. To gain a deeper insight into the role of trehalose biosynthetic proteins in crops, we assessed the enzymatic activity of the TPS/TPP family from tomato (*Solanum lycopersicum* L.) and investigated their expression pattern in different tissues as well as in response to temperature shifts. From the 10 TPS isoforms tested, only the 2 proteins belonging to class I showed enzymatic activity, while all 5 TPP isoforms investigated were catalytically active. Most of the *TPS/TPP* family members showed the highest expression in mature leaves, and promoter–reporter gene studies suggest that the two class I *TPS* genes have largely overlapping expression patterns within the vasculature, with only subtle differences in expression in fruits and flowers. The majority of tomato *TPS/TPP* genes were induced by heat stress, and individual family members also responded to cold. This suggests that trehalose biosynthetic pathway genes could play an important role during temperature stress adaptation. In summary, our study represents a further step toward the exploitation of the *TPS* and *TPP* gene families for the improvement of tomato stress resistance.

## 1. Introduction

Tomato (*Solanum lycopersicum* L.) is one of the most important vegetable crops worldwide and is grown in many areas of the globe for local use or as an export crop. It belongs to the family of Solanaceae, which includes several other commercially important species. Depending on the geographical zone, tomato is cultivated in open fields or under controlled conditions in greenhouses, and fruits are either harvested for direct consumption or for processing into a variety of end products. Tomatoes are of high nutritional value as they contain various health-promoting compounds, including vitamins, carotenoids, and phenolics [1]. Beyond its economic importance, tomato is one of the best-studied crop plants and has a long history as a model system for plant growth and developmental processes such as fruit development. The molecular tools available for tomato, including extensive genome sequence information, genetic transformation, and mutant collections, make it an ideal target for crop improvement by molecular breeding and biotechnology [2,3,4]. 

Originating in temperate climates, tomato is well adapted to almost all climatic regions of the world; however, environmental stress factors are the primary constraints of this crop’s yield potential [5,6]. Among these, abiotic stresses such as low or high temperature, deficient or excessive water, high salinity, heavy metals, and ultraviolet (UV) radiation account for a large proportion of crop yield losses [7,8].

Current predictions propose that average temperatures will rise by 3–5 °C within the next 50 to 100 years, which will dramatically affect agricultural systems on a global scale [9,10]. At the same time, the incidence of extreme weather situations such as storms, heat waves, and floods is expected to increase. In the near future, crop plants will thus encounter a greater range of stresses, occurring at a higher frequency and at the same time. Climate change, together with increasing pressure on food production due to population increase, requires intense research efforts to improve plant stress tolerance. In order to devise rational strategies for stress improvement, it is mandatory to deeply understand the cellular and molecular events during plant stress responses [11].

On the molecular level, plant stress responses involve interactions and crosstalk of many cellular pathways, including metabolite and hormonal signaling, protein kinase cascades, transcriptional reprogramming, and metabolic adjustment [11]. An important metabolic pathway that has been linked to abiotic stress tolerance in plants is trehalose biosynthesis [12,13]. Trehalose (α-*D*-glucopyranosyl-[1,1]-α-*D*-glucopyranoside) is, like sucrose, a non-reducing disaccharide widely distributed in nature [14]. The biosynthesis of trehalose involves the generation of trehalose- 6-phosphate (T6P) from glucose-6-phosphate and uridine diphosphate (UDP)-glucose by a trehalose-6-phosphate synthase (TPS) and the subsequent de-phosphorylation of T6P to trehalose and inorganic phosphate by trehalose-6-phosphate phosphatase (TPP) [15]. Historically, the presence of trehalose in plants was thought to be confined to some resurrection plants, such as *Selaginella lepidophylla*, where it might serve as a stress protectant [16]. However, research conducted during the last two decades suggests that trehalose metabolism is ubiquitous in plants, although the majority of plants accumulate only trace amounts of this sugar. This implies that trehalose likely does not act as an osmoprotectant during stress conditions in these species [12,13,17]. Heterologous expression of microbial *TPS* and *TPP* genes in plants led to improved stress resistance; however, this was frequently accompanied by profound negative effects on plant growth and development [18,19]. Further analysis revealed that changes in the level of T6P, rather than trehalose, were the cause of the phenotypes, and T6P now emerged as a central signaling molecule regulating plant growth and development, likely by integrating sucrose availability with cellular responses [17]. A direct regulatory role for T6P has been shown to exist through allosteric inhibition of the conserved energy sensor protein kinase SnRK1 [20,21]. In plants, SnRK1 responds to the availability of carbohydrates as well as to environmental stresses by down-regulating ATP-consuming biosynthetic processes while stimulating energy-generating catabolic reactions through gene expression and post-transcriptional regulation [22]. Thus, inhibition of SnRK1 by T6P couples carbohydrate availability for metabolism and growth with starvation signaling during the acclimation to various stresses [23]. However, how T6P or the proteins associated with the trehalose pathway mediate stress tolerance is far from being understood.

Sequenced plant genomes contain large families of genes encoding TPS- and TPP-related proteins [24]. The genome of the model plant *Arabidopsis thaliana* harbors 11 *TPS* genes (*AtTPS1-AtTPS11*) and 10 *TPP* (*AtTPPA-AtTPPJ*) genes [25]. The *TPS* genes can be further subdivided into class I (*AtTPS1-AtTPS4*) and class II (*AtTPS5-AtTPS11*); however, only class I encoded proteins have been shown to possess catalytic activity [26,27,28], and the role of the apparently inactive class II TPS proteins in plant metabolism remains elusive. In turn, all of the Arabidopsis TPP enzymes appear to be catalytically active, but since they have differential expression patterns, *TPP* genes probably have a tissue-, stage- and/or process-specific function [13,29]. Crop plants, such as maize, rice, or wheat, have similarly expanded TPS/TPP gene families [24].

Because of its effect on plant growth and development, modification of trehalose biosynthesis, either at the level of T6P synthesis, T6P hydrolysis, or trehalose hydrolysis, holds the potential to improve carbon partitioning and thus crop yield and biomass under various stress conditions [30,31]. Ectopic expression of the yeast *TPS1* gene in potato, tobacco, and tomato improved drought, salt, and oxidative stress resistance, providing a proof of concept for targeting the trehalose metabolic pathway to improve stress tolerance [32,33,34]. However, the pleiotropic phenotypes of these lines preclude this strategy from development into application. Interestingly, the overexpression of a bifunctional fusion of the *TPS* and *TPP* genes of *E. coli* under a stress-regulated promoter confers tolerance to salinity, drought, and cold stress in rice seedlings [35] and drought and heat stress, respectively, in tomato [36,37], largely without affecting plant phenotype under laboratory conditions [38,39]. Another promising way to circumvent unwanted effects of altered trehalose metabolism for improving abiotic stress resistance is the manipulation of endogenous trehalose metabolism-associated genes [40]. Overexpression of the enzymatically active class I *AtTPS1* in transgenic Arabidopsis rendered the lines more resistant to dehydration without affecting vegetative growth [41]. Interestingly, overexpression of the class II *OsTPS8* gene in rice conferred salinity tolerance without any yield penalty [42], suggesting that enzymatic TPS activity per se is not necessary to impart beneficial effects on plant stress tolerance. A number of *TPP* genes have also been associated with abiotic stress responses. Overexpression of *AtTPPI* or *AtTPPF* enhanced plant drought tolerance [43,44]. Nuccio et al. [45] overexpressed the rice *OsTPP1* gene under a spikelet-specific promoter in transgenic maize plants. Developing ears displayed reduced T6P content and increased sucrose levels compared with non-transgenic plants, suggesting an improved sink function of the reproductive tissues. Data from field experiments indicated that transgenic maize plants showed increases in both kernel set and harvest index under a range of environmental conditions. Importantly, yield under drought conditions was improved by 31% to 123% under severe drought conditions, relative to yields from non-transgenic controls [45].

Even though the complement of trehalose biosynthesis genes has been identified in the tomato genome sequence and a limited expression study has been conducted [46], we are still far from a thorough understanding of the roles and effects that trehalose/T6P has on tomato growth and development as well as on stress responses. Towards this goal, we investigated the functionality of the proteins encoded by the tomato trehalose biosynthesis genes through yeast complementation. In addition, the differential expression of the genes in response to temperature was analyzed, and high-resolution expression analysis of the two enzymatically active class I TPS proteins was conducted using promoter–reporter gene studies in stably transformed tomato lines.

## 2. Results

### 2.1. Catalytic Activity of S. lycopersicum Class I TPS Proteins SlTPS2 and SlTPS8

A previous survey of the tomato genome identified 11 potential TPS encoding genes that have been designated *TPS1* to *TPS11* based on their chromosomal location ([46]; Appendix A). However, the *SlTPS11* has been predicted to encode a pseudogene [46], similar to *At*TPS3 in Arabidopsis [27]. A phylogenetic comparison of predicted tomato TPS protein sequences to the TPS complement found in Arabidopsis suggests the existence of two clades of class I proteins and four clades of class II isoforms (Figure 1a). Arabidopsis contains three enzymatically active class I TPS proteins (*At*TPS1, *At*TPS2, *At*TPS4) [28], and two tomato TPS isoforms (*Sl*TPS2 and *Sl*TPS3) cluster most closely together with *At*TPS1; thus, likely representing the catalytically active tomato class I isoforms (Figure 1a). Similar to the structure of *At*TPS1, *Sl*TPS2 and *Sl*TPS8 both contain N- and C-terminal domains flanking a central glycosyltransferase domain that contains the catalytic site and has similarity with single-domain TPS enzymes in bacteria (Figure 1b). The N-terminal domain has been shown to have an auto-inhibitory effect on the enzyme’s activity when expressed in yeast [47]. In addition, almost all of the amino acid residues critical for enzymatic function are conserved between Arabidopsis and tomato (Figure 1b). This includes *At*TPS1 L27 (L42 in *Sl*TPS2 and L26 in *Sl*TPS8), required for the auto-inhibitory function of the N-terminal extension [47], a potential phosphorylation site at S252 (S268 in *Sl*TPS2 and S251 in *Sl*TPS8) [48], and the catalytic triad at R369/K374/E476 (R385/K390/E492 in *Sl*TPS2 and R368/K373/E475 in *Sl*TPS8) [49]. An exception is the alanine residue at position 119 of *At*TPS1, which has been suggested to play an important catalytic function [27]. While *Sl*TPS8 carries a conservative glycine substitution at this position (G118), *Sl*TPS2 features a valine (V125). However, similar to alanine in *At*TPS1, both substitutions are non-polar residues and thus likely will have no disruptive effect on enzymatic activity. The C-terminal domain has a similarity with plant TPP enzymes but lacks some of the residues associated with the active site of TPP enzymes and is thus presumed to have no catalytic activity [24]. As in Arabidopsis, tomato class II TPS proteins also consist of a glucosyltransferase domain as well as a C-terminal TPP-like domain; however, they generally lack the conserved residues required for catalytic activity (Figure 1b).

In order to verify the catalytic activity of the class I tomato TPS proteins and to exclude any cryptic enzymatic activity in the tomato class II proteins, we tested their ability to complement the growth of a yeast mutant. The *Δtps1* mutant in the W303 background lacks endogenous TPS activity, and yeast cells are unable to grow on glucose as the sole carbon source because of an uncontrolled glycolytic flux leading to ATP depletion [47]. The coding sequences of all class I and class II tomato TPS proteins, except TPS11, were PCR amplified from cDNA prepared from a tomato tissue mixture and cloned into the yeast expression vector pYX212. Expression of the TPS sequences in this vector is controlled by the constitutive *HXT7* promoter, and proteins carry a C-terminal HA-tag to facilitate immunodetection. After transformation into the *Δtps1* mutant, all strains grew equally well on galactose (Figure 2B). The *Δtps1* mutant grows normally on galactose because the influx of galactose into glycolysis is not regulated by T6P. The growth phenotype on glucose was readily complemented by ectopic expression of the yeast *TPS1* gene, indicating the validity of the approach. However, only the two class I TPS proteins, *Sl*TPS2 and *Sl*TPS8, were able to complement the growth defect on glucose, while none of the class II proteins showed this ability (Figure 2B). The expression of all proteins was confirmed by immunoblotting (Figure 2C). Interestingly, complementation with *Sl*TPS8 conferred robust yeast growth, while considerably lower growth was observed in the case of *Sl*TPS2, despite both proteins accumulating to similar levels.

Growth assays in liquid medium confirmed that expression of *Sl*TPS2 and *Sl*TPS8 restored growth of the *Δtps1* mutant on glucose (Figure 3). Interestingly, in this case, *Sl*TPS2 almost restored wild-type growth levels, while the *Sl*TPS8 complemented strain grew considerably slower over the 48 h period of the experiment. Again, none of the tomato class II TPS proteins was able to complement the *Δtps1* mutant growth phenotype in liquid medium with glucose as the sole carbon source (Figure 3).

Collectively, our data suggest that the tomato genome encodes two class I TPS proteins, both displaying enzymatic activity.

### 2.2. All Tested Tomato TPP Enzymes Have Catalytic Activity in Yeast

The tomato genome encodes eight predicted *TPP* genes, and the expression of four of these (*SlTPP2*, *SlTPP3*, *SlTPP4*, and *SlTPP8*) is supported by available ESTs [46]. A phylogenetic tree suggests that many of the tomato TPP proteins occur in duplicates (Figure 4), similar to what has been found in Arabidopsis and which has been interpreted as an expansion of the gene family as a result of genome duplication [29]. However, it is not clear whether all members have retained enzyme activity. The yeast *Δtps2* mutant lacks endogenous TPP activity and exhibits a thermosensitive growth phenotype at 39 °C due to the accumulation of high levels of T6P [50]. We cloned the full-length coding sequences of *Sl*TPP1, *Sl*TPP2, *Sl*TPP3, *Sl*TPP4, and *Sl*TPP8 into the pYX212 yeast expression vector as above. Despite extensive efforts, we were not able to PCR amplify the coding sequences of *Sl*TPP5, *Sl*TPP6, and *Sl*TPP7 from any tomato tissue tested, well in line with the absence of corresponding ESTs in the database [46]. Thus, these genes are either not expressed, or expression is confined to specialized cell types not captured by our sampling approach. When transformed into yeast, all tested tomato TPP isoforms were able to complement the yeast *Δtps2* high-temperature growth phenotype to a similar extent (Figure 5B), indicating they all possess TPP enzymatic activity. Comparable expression of all proteins investigated was verified by immunoblotting with an anti-HA antibody (Figure 5C).

### 2.3. Tissue-Specific Expression Patterns of TPS and TPP Genes in Tomato

Real-time quantitative PCR (qPCR) measurements were performed in order to analyze the expression of *Sl*TPS and *Sl*TPP gene expression in different organs and during plant development (Figure 6a). The expression heatmap (Figure 6b) shows that the majority of tomato *TPS/TPP* genes have their peak expression in mature source leaves. Notable exceptions are the two class I TPS genes, with *Sl*TPS2 displaying the highest expression in flowers and *Sl*TPS8 showing its strongest signal in orange fruits, followed by flowers (Figure 6b). *Sl*TPP1 and *Sl*TPP4 have their highest expression in roots, and the latter isoform shows hardly any expression in other tissues tested.

With the exception of orange fruits, class I *Sl*TPS2 and *Sl*TPS8 show largely overlapping expression patterns, which begs the question of whether they are functionally redundant or the resolution of the qPCR analysis is simply not high enough to reveal further differences. To further analyze the spatial expression patterns of these genes, the 1.5 kb region upstream of the translational start site (Appendix A), likely representing the respective gene promoter including the 5’ UTR, was PCR amplified, fused to the β-glucuronidase (GUS) reporter gene and transformed into tomato to generate stable transgenic lines. The analysis of several independent transgenic lines for each promoter–reporter gene construct largely confirmed the overlapping expression pattern of both genes (Figure 7). However, there were subtle differences in the GUS staining pattern on the cell-type-specific level. While *pSlTPS8::GUS* staining was mainly observed in the pericarp of orange fruits, *pSlTPS2::GUS* driven reporter gene activity is constrained to the seeds (Figure 7A,K). In green fruits, GUS staining was only observed within the vasculature for both constructs (Figure 7B,L). A slight difference in expression between both constructs could also be detected in flowers (Figure 7F,P). Both constructs mediated GUS activity within the style immediately in the region below the stigma. However, *pSlTPS2::GUS* yielded a confined signal in a specific cell layer in the stamen, *pSlTPS8::GUS* showed more diffuse expression in this region. In leaves, both reporter-gene constructs led to the strongest GUS staining in the vasculature (Figure 7G,Q). Strong GUS staining of the vasculature was also observed in stems (Figure 7I,S). The differentiation zone of roots was strongly stained in *pSlTPS2::GUS* plants, while GUS staining of roots was considerably weaker in *pSlTPS8::GUS* plants (Figure 7J,T).

In summary, the data suggest that TPS and TPP genes not only display distinct differences in their tissue-specific expression patterns but also might exhibit cell-type-specific expression in certain tissues.

### 2.4. Identification of Temperature-Responsive SlTPS and SlTPP Genes 

To assess the responsiveness of trehalose metabolism genes to temperature, 4-week-old tomato plants were either kept under control conditions (25 °C), subjected to heat (40 °C) or chilling (5 °C) stress, or submitted to suboptimal growth temperatures (15 °C), each over a time course of 60 h. At different time points, mature leaves were sampled, and the expression of trehalose genes was analyzed by qPCR. The expression heatmap in Figure 8 shows that the majority of genes tested display strong induction upon heat stress, mostly peaking at the 12 h time point. A marked difference occurred between the two class I TPS isoforms, with a strong induction of *Sl*TPS2 after 12 h of heat and only a slight increase in *Sl*TPS8 at the same time point. However, *Sl*TPS8 expression was induced late during chilling stress (60 h), where *Sl*TPS2 showed no response. *Sl*TPP1 was the sole member of the TPP family induced during cold stress. Suboptimal temperatures of 15 °C had no marked effect on gene expression with the exception of *Sl*TPP8, which showed a transient induction after 12 h (Figure 8).

Collectively, the expression analysis suggests that tomato *TPS* and *TPP* genes are highly responsive to changes in temperature and might play a role in the adaption to heat stress in particular.

## 3. Discussion

In this study, we present the initial functional characterization of the *TPS* and *TPP* family of the important vegetable crop tomato. Both proteins together constitute the principle pathway of trehalose biosynthesis in plants [15]. The pathway, and especially its intermediate T6P, play an essential role in the regulation of plant growth and development [17,51,52] and have been implicated in plant adaptation to a range of abiotic stress conditions [13,30]. Thus, the trehalose biosynthetic pathway and its associated gene products are promising targets to improve plant performance under stress. In tomato, the *TPS* and *TPP* gene families show a similar expansion as has been described for other plants [24,29,46]. The phylogenetic analyses suggest that 2 of the 10 expressed *TPS* genes encode for class I TPS proteins, which both possess enzymatic activity based on their ability to complement the yeast *Δtps1* mutant. This is in contrast to the model plant Arabidopsis, which encodes three enzymatically active class I TPS proteins, *At*TPS1, *At*TPS2, and *At*TPS4 [28]. However, Arabidopsis seems to be exceptional in this respect for a number of reasons. *At*TPS2 and *At*TPS4 lack the N-terminal auto-inhibitory region found in *At*TPS1, and these short forms of TPS appear to be confined to the Brassicaceae [24]. Accordingly, it was previously assumed that most plants encode a single class I TPS [24]; however, the recent refinement of available genome information suggests that many plants encode 2 or 3 class I TPS proteins, which seem to possess the N-terminal auto-inhibitory domain [53,54,55,56,57,58]. However, the enzymatic activity of these TPSs has only been confirmed for the two class I isoforms of the moss *Physcomitrella patens* [58]. Curiously, *Sl*TPS2 showed a reduced ability to complement the yeast *Δtps1* mutant in plate assays relative to *Sl*TPS8, while in liquid growth assays, the situation was reversed. This was consistently observed over several repetitions of the experiments. Both proteins appear to accumulate to similar levels, and thus the phenomenon cannot be simply explained by differences in expression levels. Although we cannot rule out experimental or technical artifacts, this observation might reflect some subtle differences in the biochemical properties between the two isoforms that warrant further investigation. 

The tomato class I TPS isoforms *Sl*TPS2 and *Sl*TPS8 cluster together with *At*TPS1 in the phylogenetic tree, and both possess an N-terminal domain that appears slightly extended in *Sl*TPS2. In Arabidopsis, *At*TPS1 is assumed to constitute the majority of in planta TPS activity, and the *tps1* loss-of-function mutant shows severe developmental defects, including embryo lethality [59,60]. A phenotype of the loss of function of *tps2* and *tps4* has not been described in Arabidopsis. Thus, the Arabidopsis class I TPS enzymes appear not to be functionally redundant. In addition to the differences in domain architecture, this is also reflected by fundamental differences in their expression patterns. In contrast to *AtTPS1*, which is expressed in all major organs of the plant, expression of *AtTPS2* and *AtTPS4* is largely restricted to specific tissues within developing seeds [27,52]. This is in contrast to *SlTPS2* and *SlTPS8*, which show largely overlapping expression patterns with the exception of orange fruits. Thus, the question arises whether both isoforms have overlapping functions in most tissues and, if so, why tomato expresses two active TPS proteins. The expression of *SlTPS8* in fruits might point to a specialized function of this isoform in tomato fruit development. Interestingly, grape also has two class I TPS proteins, one of which appears to be preferentially expressed in fruits [56]. The slight differences in expression within flower tissues further support the view of a specialized function of *Sl*TPS2 versus *Sl*TPS8 in certain tissue or cell types. The promoter-GUS stainings suggest that *Sl*TPS2 and *Sl*TPS8 are predominantly expressed in the vasculature of leaves, stems, and developing fruits. This is similar to *At*TPS1, which is primarily expressed in and around the vascular tissue around the phloem parenchyma of the bundle sheath [61]. Given the assumption that T6P is an integrator of sucrose levels [62], a model has been put forward in which a rise in phloem sucrose levels due to a reduction in sink demand will trigger the synthesis of T6P in the phloem parenchyma. T6P will be able to diffuse symplastically into the mesophyll cells, where it can divert photoassimilates away from sucrose toward organic and amino acids during the day or slow down the remobilization of transitory starch reserves at night [17]. Thus, TPS activity would play an important role in balancing sucrose production and export in source tissues with carbohydrate consumption in sink tissues. Given the similarities between the expression patterns of *At*TPS1 and class I enzymes in tomato, *Sl*TPS2 and *Sl*TPS8 might serve similar functions in metabolic regulation.

In addition to their already important function during growth under normal conditions, TPS enzymes might become particularly important under stress conditions. Both *SlTPS2* and *SlTPS8* are induced during heat stress, albeit to a different extent and following a different time course. In addition, *SlTPS8* expression is induced during chilling stress, pointing toward a particular role of this isoform in low-temperature adaption. Although for these hypotheses to be tested, additional experiments are required, including genetic approaches, it appears likely that *Sl*TPS2 and *Sl*TPS8 serve overlapping functions during vegetative growth under ambient conditions but become functionally specialized in certain tissues and cell types or under certain stress conditions.

None of the class II tomato TPS isoforms possess apparent catalytic activity based on their failure to complement the growth of the yeast *Δtps1* mutant on glucose. This is in line with the data obtained from the analysis of the Arabidopsis class II TPS proteins, which also do not show catalytic activity in a reproducible fashion [27,63]. In the absence of any demonstrable enzymatic activity, the function of class II TPS proteins remains elusive. However, loss-of-function and overexpression studies have implicated class II TPS in the adaptation to a range of environmental stimuli, including thermotolerance and cold tolerance [64,65]. Our qPCR data suggest that under ambient conditions, class II *TPS* genes have their highest expression in mature source leaves, and many seem to be upregulated upon heat stress. Whether this expression pattern is functionally relevant for the plant to adapt to heat needs to be clarified by future experiments. The molecular mechanisms through which class II TPS proteins could be integrated into cellular processes, including stress signaling, is unclear. Previous data from rice suggest that class I and class II TPS proteins form intricate protein–protein interaction networks among each other, potentially having a regulatory impact on enzymatically active TPS enzymes [66]. Furthermore, given the conservation of the respective binding sites, it has been suggested that class II enzymes could act as signaling proteins binding small molecules or metabolites such as T6P [24]. Future experiments have to address the functional relevance of heat induction of TPS gene expression in tomato using genetic approaches such as CRISPR genome editing or transgenic overexpression.

From the 8 *TPP* genes identified in the tomato genome [46], we were able to clone 5 by PCR from mixed tissue samples. It has been previously reported that for *SlTPP1*, *SlTPP5*, *SlTPP6*, and *SlTPP7,* no ESTs or full-length cDNAs were available, and thus it is not clear whether these are expressed at all [46]. This fact notwithstanding, we were able to isolate the *SlTPP1* cDNA in our experiments, providing proof for its expression and suggesting that the other missing isoforms might be expressed in a narrow range of tissues or cell types not easily captured by our sampling strategy. The number of predicted *TPP* genes in tomato is comparable to what has been reported for Arabidopsis and similar to the situation in this model plant, a number of tomato TPP proteins form paralogous pairs indicating a similar evolutionary trajectory of the family in both plants, including their expansion by whole-genome duplication [27]. It has been postulated that the strong genome duplication bias in the history of the TPP gene family hints at a potential regulatory role for the TPP proteins [27]. Apparently, all of the 5 tomato TPP proteins tested have conserved their enzymatic function during evolution which, in conjunction with their specific spatiotemporal expression patterns, expands the possibility for the plant to adjust T6P levels according to specific needs or environmental situations, such as temperature stress. It should be noted that regulatory functions of individual TPP isoforms in plant development might be independent of their enzymatic activity [67], thus further expanding the range of possible roles of this protein family in the regulation of cellular processes. Future experiments will have to investigate the extent to which neo-functionalization occurred within the tomato TPP family by using a combination of genetic, biochemical, and cell biology approaches.

## 4. Materials and Methods

### 4.1. Plant Material, Growth Conditions, and Stress Treatments

All plants were cultivated in Grossbeeren, Germany (52°20′55.6″ N 13°18′39.0″ E). Tomato plants (*Solanum lycopersicum* cv. Moneymaker) were grown in pots in a mixture of soil (80%) and sand (20%) in a climate chamber with daily watering and subjected to a 16-h light:8-h dark cycle (25 °C:20 °C) with 240–300 µmol m^−2^ s^−1^ light (Metal Halide Lamp MT400DL/BH; Iwasaki Electric Co., Tokyo, Japan). For the temperature stress experiment, 4-week old-plants were divided into 4 groups and subjected to normal (25 °C), heat (40 °C), cold (4 °C), and suboptimal (15 °C) temperatures with three replications. The samples were harvested from mature leaves at three different time points; 0 h, 12 h, and 60 h after reaching the desired temperature and immediately frozen in liquid nitrogen and stored at −80 °C until further analysis. 

### 4.2. Plasmid Construction

The full-length coding region of each tomato TPS or TPP gene was amplified from cDNA by PCR and inserted into the yeast multicopy pYX212 plasmid with an *HXT7* promoter and *URA3* marker, without stop codon and in frame with C-terminal double hemagglutinin (HA) tag using the NEBuilder DNA assembly kit (New England Biolabs, Ipswich, MA, USA). Positive clones were identified by restriction digest and verified via sequencing. For GUS reporter constructs, the 1.5 kb region immediately upstream of the predicted *SlTPS2* and *SlTPS8* translational start site was amplified by PCR using tomato genomic DNA as a template. The resulting fragments were inserted into the pENTR-D/TOPO cloning vector (Thermo, Waltham, MA, USA), sequence verified, and subsequently mobilized into the pK7GWIWG2 vector [68] using L/R-recombination (Thermo, Waltham, MA, USA). Oligonucleotides used for cloning are listed in Appendix A.

### 4.3. Yeast Complementation

For the yeast (*Saccharomyces cerevisiae*) growth complementation assay, the yeast W303-1A wild-type strain (*Matα leu2-3, 112 ura3-1trp1-1his3-11,15 ade2- 1 can1-100 GAL SUC2*), the *tps1Δ* deletion strain YSH290 (*W303-1A, tps1Δ:: TRP1*), and the *tps2Δ* deletion strain YSH448 (*W303-1A, tps2Δ::HIS3*; [69]) were used (kindly provided by Fillip Rolland, KU Leuven). Yeast transformation was performed using the LiAc-method [70]. Transformants were selected on medium without uracil. Plates were incubated at 30 °C (for *tps1Δ* complementation) or at 39 °C (for *tps2Δ* complementation). Colony growth was monitored after 48 h. Growth curves were started in liquid medium in 96-well plates at an OD600 of 0.05. Growth was monitored by OD600 measurement every 1 h with continuous agitation at 30 °C in an Infinite M200 Pro plate reader (Tecan, Männedorf, Switzerland) for 2 days.

### 4.4. Immuno-Detection of Protein Expression in Yeast

A total of 5 mL yeast cell cultures with an OD600 = 1 were harvested and resuspended in ice-cold lysis buffer (20 mM Tris pH 7.6, 2% Triton X-100, 100 mM NaCl, 1 mM EDTA, pH 8; 1% SDS) containing protease inhibitors (complete EDTA-free, Roche, Basel, Switzerland). Cells were lysed by vortexing (0.5-mm glass beads). After centrifugation at 4 °C for 10 min at maximum speed, protein concentrations in the supernatant were determined using the dye-binding assay (BioRad, Hercules, CA, USA) and equal protein amounts of the soluble extract were subjected to SDS gel electrophoresis followed by electrotransfer of proteins to a nitrocellulose membrane. HA-tagged proteins were decorated with an anti-HA antibody (1:500; Merck; cat. no. 12013819001) and detected using the Clarity Western ECL substrate (BioRad). Chemiluminescence was detected using a ChemiDoc imager (BioRad, Hercules, CA, USA).

### 4.5. RNA Extraction and RT-qPCR

Total RNA was isolated from the respective tomato tissue and then treated with RNase-free DNase to degrade any remaining DNA. RNA concentrations were measured using a microplate reader (Tecan). For RT-qPCR, first-strand cDNA synthesis was performed from 1 µg of total RNA using Revert-Aid reverse transcriptase (Thermo), and the cDNAs were amplified using SensiFAST SYBR Lo-ROX Mix (Bioline, London, UK) in the AriaMx Realtime PCR System (Agilent Technologies, Santa Clara, CA, USA) as previously described [71]. At least three biological and two technical replicates were used for each analysis. The transcript level was standardized based on cDNA amplification of the ubiquitin gene for reference. Primer sequences are provided in Appendix A.

### 4.6. Generation of Transgenic Tomato Plants

The promoter–reporter gene constructs were transformed into *Agrobacterium tumefaciens* CV58C1, carrying the hypervirulence attenuated tumor-inducing helper plasmid pGV2260. Tomato cotyledon explants (cv. Moneymaker) were transformed and regenerated as described by Ling et al. [72].

### 4.7. Analysis of GUS Reporter Lines

Different developmental stages of at least four independent transgenic *pSlTPS2::GUS* and *pSlTPS8::GUS* lines were investigated for tissue-specific localization patterns using histochemical staining according to Jefferson et al. [73]. In brief, plant samples were submerged for 20 min in ice-cold 90% acetone. The acetone was then removed by replacing it with 2 mL of buffer A (50 mM Na_2_HPO_4_, 10 mM K_4_[Fe(CN)_6_], 10 mM K_3_[Fe(CN)_6_], 0.2% Triton X-100, pH 7.0) and vacuum infiltrated with GUS staining solution (buffer A plus 2 mM (f.c.) µL X-Gluc (5-bromo-4-chloro-3-indolyl-β-glucuronic acid) solution in DMSO) three times. Subsequently, the samples were incubated at 37 °C overnight. On the next day, the GUS staining buffer was carefully removed and replaced with 70% ethanol to bleach the green chlorophyll. Several changes of ethanol were performed until the tissue turned white. A stereo microscope imaging system (Discovery V20; Zeiss) was used to collect the images. The same settings were used to image a particular tissue from different plants. The entire experiment was carried out twice with independent batches of plants.

## Figures and Tables

**Figure 1 ijms-23-11436-f001:**
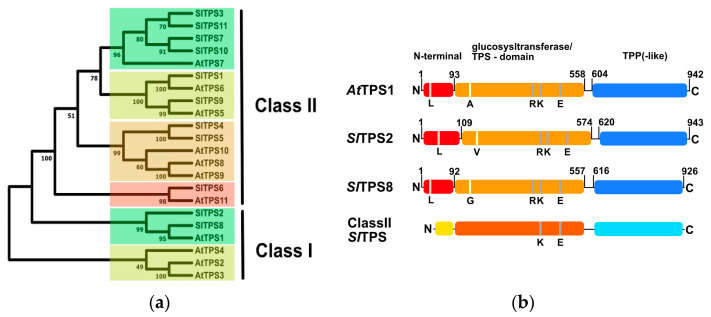
Comparison of the TPS families of tomato and Arabidopsis. (**a**), Phylogenetic tree of the TPS-families from tomato and Arabidopsis inferred using the Neighbor-Joining method. The bootstrap consensus tree inferred from 500 replicates. Evolutionary analyses were conducted in MEGA11 (www.megasoftware.net, accessed on 24 July 2022) (**b**), Domain structure of tomato class I TPS proteins in relation to Arabidopsis TPS1. For comparison, the general domain structure of tomato class II TPS proteins is indicated. Domains were predicted using the InterPro online tool (https://www.ebi.ac.uk/interpro/, accessed on 1 August 2022).

**Figure 2 ijms-23-11436-f002:**
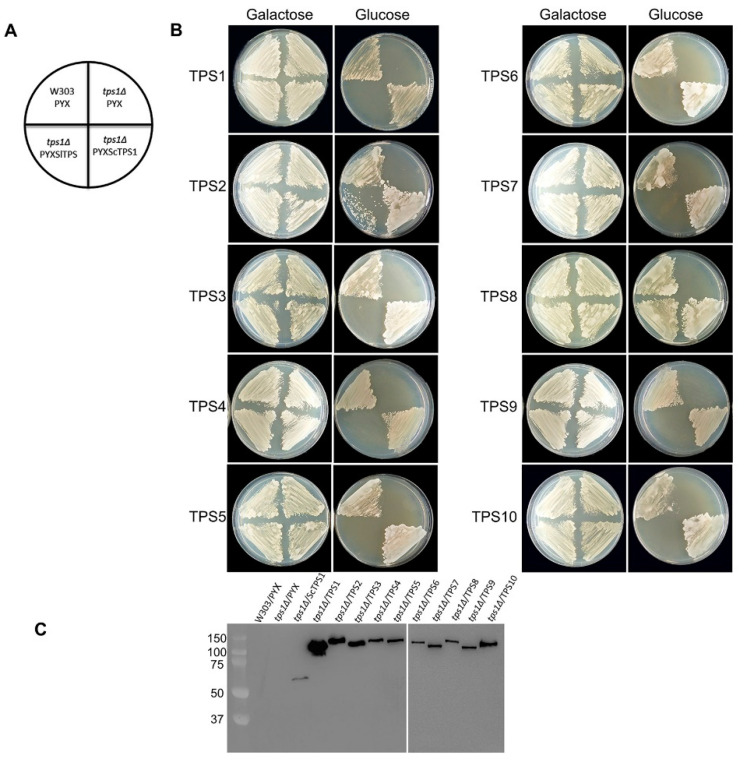
*Sl*TPS2 and *Sl*TPS8 display heterologous trehalose-6-P synthase (TPS) activity when grown on glucose plates. (**A**) Layout of growth plates. (**B**) Complementation of the yeast *Δtps1* growth defect on glucose. Growth on galactose served as a control. (**C**) Verification of protein expression in the individual yeast strains by immunoblotting using an anti-HA antibody. The entire experiment has been repeated three times using three independent plates, each with similar results. A representative image is shown here.

**Figure 3 ijms-23-11436-f003:**
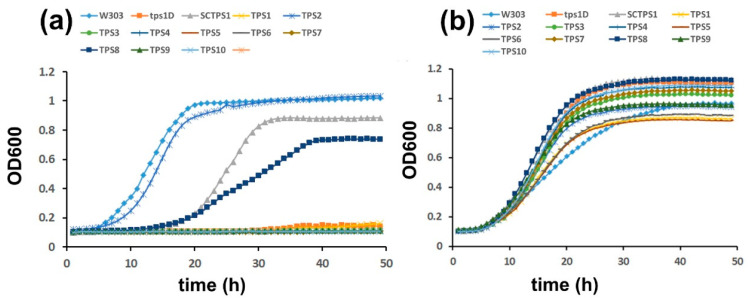
*Sl*TPS2 and *Sl*TPS8 display heterologous trehalose-6-P synthase (TPS) activity when grown in liquid culture. For growth curves, pre-cultures were diluted to OD600 = 0.05 and subsequent growth was followed by monitoring OD600 for 48 h. (**a**), Growth on glucose (**b**), Galactose control. The experiment was carried out three times with similar results. A representative data set is shown.

**Figure 4 ijms-23-11436-f004:**
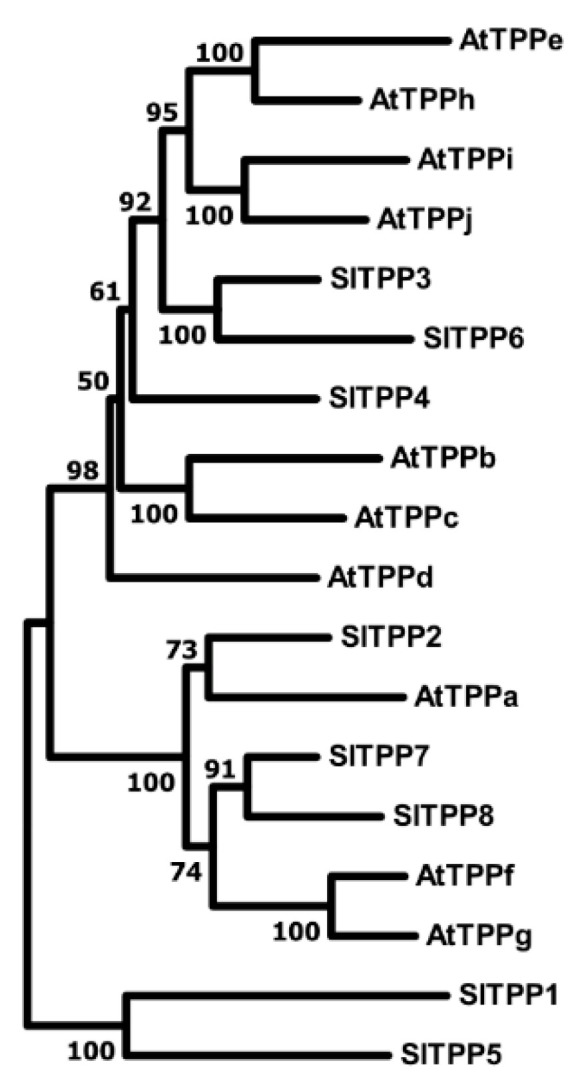
Phylogenetic tree of the TPP-families from tomato and Arabidopsis inferred using the Neighbor-Joining method. The bootstrap consensus tree inferred from 500 replicates. Evolutionary analyses were conducted in MEGA11 (www.megasoftware.net, accessed on 24 July 2022).

**Figure 5 ijms-23-11436-f005:**
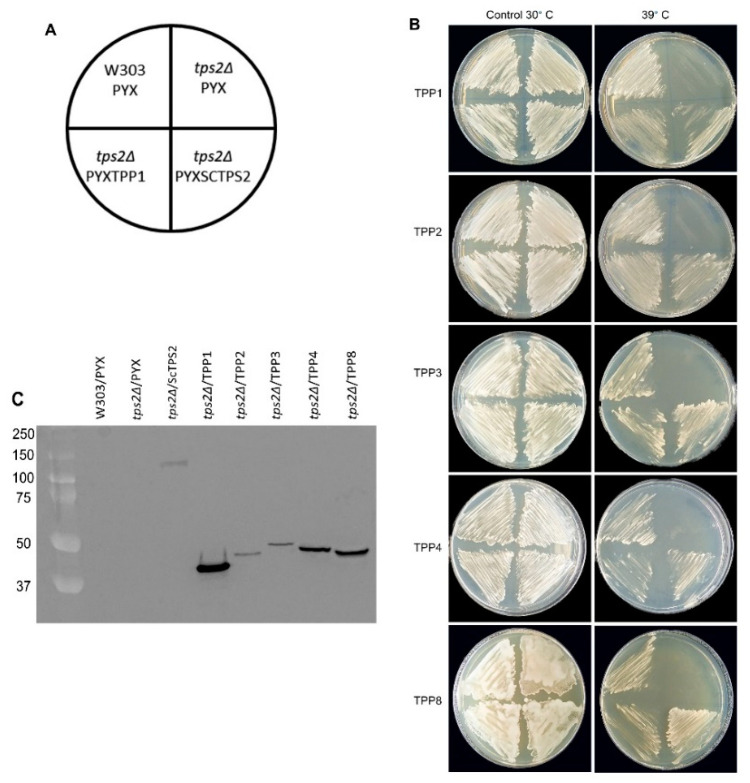
All tomato TPP isoforms tested display heterologous trehalose-6-P phosphatase (TPP). (**A**) Layout of growth plates. (**B**) Complementation of the yeast *Δtps2* growth defect at 39 °C. Growth at 30 °C served as a control. (**C**) Verification of protein expression in the individual yeast strains by immunoblotting using an anti-HA antibody. The experiment has been repeated three times with three independent plates each, yielding similar results. Representative images are shown here.

**Figure 6 ijms-23-11436-f006:**
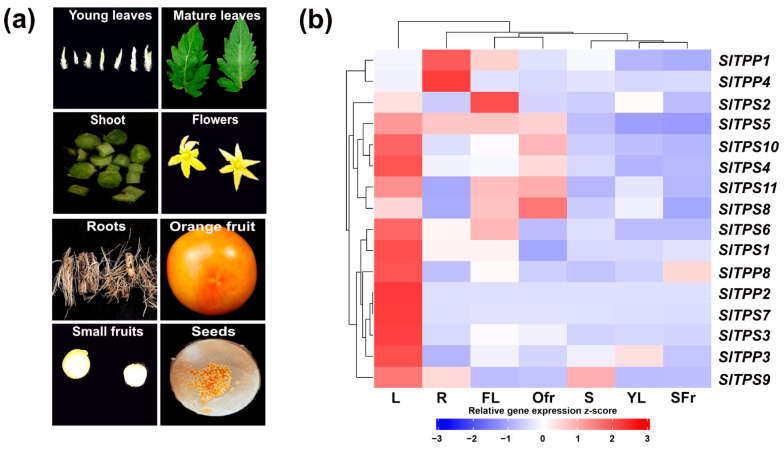
Tissue-specific expression of tomato *TPS/TPP* genes. (**a**) Representation of tissues sampled for RNA extraction. (**b**) Heatmap of gene expression. L = mature leave; R = root; FL = flower; Ofr = orange fruit; S = stem; YL = young leave; SFr = small fruit. The experiment was carried out at least three times with similar results.

**Figure 7 ijms-23-11436-f007:**
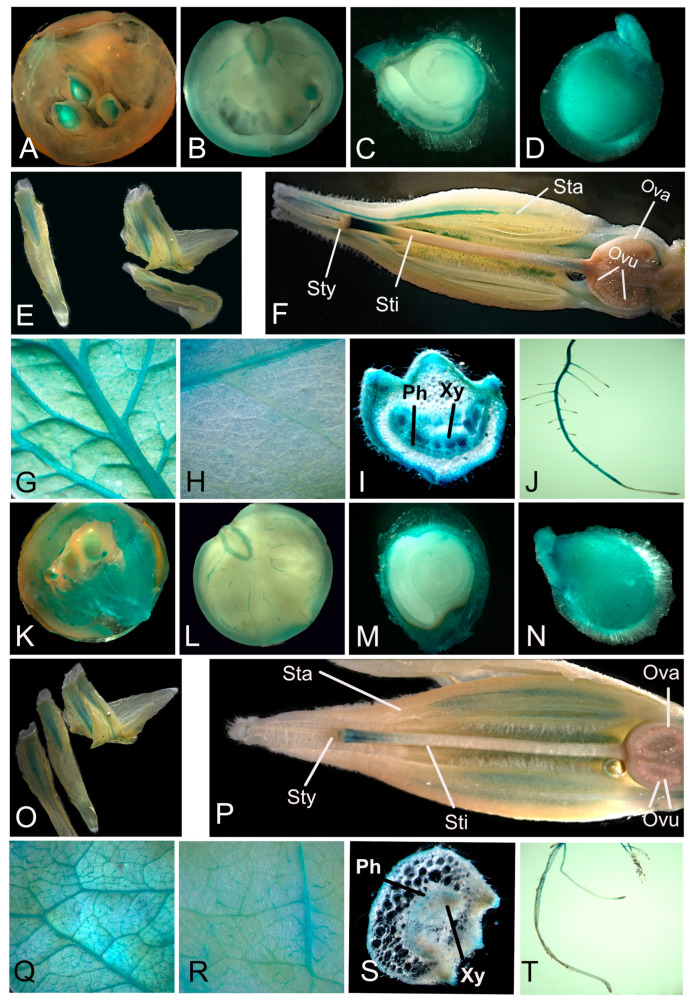
Promoter—GUS analysis of *SlTPS2* and *SlTPS8* expression (**A**–**J**) *pSlTPS2::GUS*. (**K**–**T**) *pSlTPS8::GUS.* Sty = style; Sti = stigma; Sta = stamen; Ova = ovarium; Ovu = ovule; Ph = phloem; XY = xylem. For each tissue, at least four independent transgenic lines were analyzed. The experiment was carried out twice with independent batches of plants and representative pictures are shown here.

**Figure 8 ijms-23-11436-f008:**
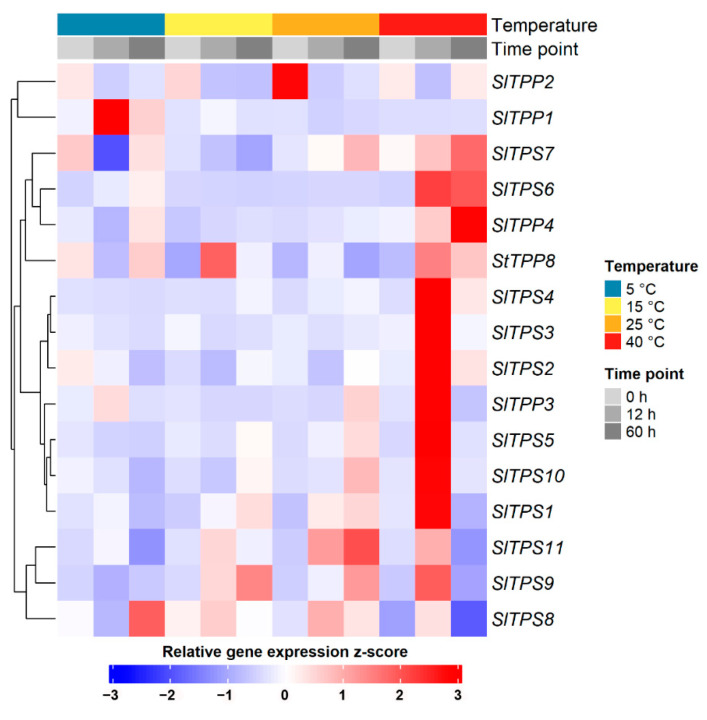
Temperature-dependent changes in expression of tomato *TPS and TPP* genes. The experiment was carried out three times with similar results.

## Data Availability

All raw data inherent to the reported findings are available from the corresponding author upon request.

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
