# Peer review of "Characterization of Trehalose-6-Phosphate Synthase and Trehalose-6-Phosphate Phosphatase Genes of Tomato (Solanum lycopersicum L.) and Analysis of Their Differential Expression in Response to Temperature"

_ijms, 2022, doi:10.3390/ijms231911436_

Round 1

Reviewer 1 Report

This is a straightforward characterization of two gene families involved in trehalose biosynthesis/regulation in tomato. The study conducts phylogenetic analyses, functional (complementation) assays, and expression analyses on the trehalose 6-phosphate synthase (TPS) and the trehalose-6-phosphate phosphatase (TPP) genes of tomato. The results are similar to those from other organisms but do provide new information regarding the tomato genes in particular and also serve to support emerging models and trends describing this gene family in plants overall.

The existing knowledge on the role of trehalose and the TPS/TPP genes is summarized adequately, the results are shown and explained clearly, and the conclusions are supported by the data. All of the experiments appear to have been conducted correctly and with appropriate controls (with the possible exception of the GUS staining, see below). Data analysis and statistics appear to be correct.

Figure 1 is missing the “I” and “II” after the bolded “Class” for the phylogeny.  I also suggest adding protein diagrams for the remaining TPS genes, which may help AtTPS1, SlTPS2, and SlTPS8 appear differentiated. This is only a suggestion as I think it would be interesting to see all of the TPS genes and predicted domains.  Also, please detail how the domains were identified in the figure legend or methods section.

Please attempt some explanation of why your TPS2 showed greater activity in liquid media while TPS8 was greater on plates.  How consistent was this? It does not impact the interpretation of your findings too much but is very curious. Is the image of the TPS2 plate representative of multiple trials (plates)? There is always some variation in yeast growth like this, could the lower growth with TPS2 just be from random noise of the assay? Or was this finding repeatable?

For GUS assays it is more typical to show the same tissues alongside different targets or controls for comparisons.  Are the images shown representative of multiple assays for each tissue/target? Would it be possible to redesign this figure to show representative images from TPS2, TPS8, and negative controls for each tissue type such that they were side by side instead of all grouped by gene and with no controls shown? I’m not sure that you can make claims comparing TPS2 and TPS8 expression based on the shown images alone.  The qPCR data is more convincing.  For instance, Figure 7J and 7T seem to show major differences in TPS2 vs TPS8 but the qPCR shows both genes high in roots. Could the apparent difference be due to camera setting (exposure time) or something? Is there a way to show how consistent these GUS experiments were from replicate to replicate? 

Minor considerations:

The first half of the first paragraph could be omitted as knowledge of what a tomato is and what it is used for can be assumed.

Line 55: I suggest that you replace “will increase” with “are expected to increase”.

Line 253: I believe this should be 5’ UTR (not 3’ UTR).

Line 460: More detail is needed for the GUS methodology. How many replicates were imaged? What type of instrument and settings were used? What kind of controls?

Author Response

First of all, we would like to thank the reviewer for their constructive comments. Find a point-by-point response to the remarks below.

Figure 1 is missing the “I” and “II” after the bolded “Class” for the phylogeny.  I also suggest adding protein diagrams for the remaining TPS genes, which may help AtTPS1, SlTPS2, and SlTPS8 appear differentiated. This is only a suggestion as I think it would be interesting to see all of the TPS genes and predicted domains.  Also, please detail how the domains were identified in the figure legend or methods section.

Response: The “I” and “II” were visible in the original submission but appear to be altered during handling. We have adjusted Figure 1(a) again to correct this error. The domain structure of tomato class II TPS proteins is the same across all members. Thus, a schematic representation of all class II proteins would be repetitive and does not add additional information. However, we agree with the reviewer that a general comparison of the class I enzymes with class II proteins would be helpful. Thus, we added a schematic representation of the class II protein domain structure to Figure 1B. The conserved domains in the TPS proteins were predicted using the InterPro online tool (https://www.ebi.ac.uk/interpro/). This information has been added to the figure legend.

Please attempt some explanation of why your TPS2 showed greater activity in liquid media while TPS8 was greater on plates.  How consistent was this? It does not impact the interpretation of your findings too much but is very curious. Is the image of the TPS2 plate representative of multiple trials (plates)? There is always some variation in yeast growth like this, could the lower growth with TPS2 just be from random noise of the assay? Or was this finding repeatable?

Response: this is indeed an interesting point. The observation is highly reproducible as it consistently appeared in all three repetitions of the experiment (with three independent plates each in the plate assays). We acknowledge this fact in the discussion section of the revised version of the ms. However, we currently have no good explanation for this observation although it might suggest subtle differences in the biochemical properties between TPS2 and TPS8, which in turn could point to isoform specific biological functions. Further experiments on the biochemical as well as on the genetic level are required to further elucidate this issue.

For GUS assays it is more typical to show the same tissues alongside different targets or controls for comparisons.

Response: we are not sure if we understand the reviewer’s remark. The wild type control does contain GUS activity and therefore does not show staining. Since we do not apply a particular treatment to the GUS plants to induce expression, there’s actually nothing to compare to, except the comparison between SlTPS2 and SlTPS8. Thus, we feel it does not add additional information when pictures of unstained wild type tissue are incorporated into the figure.

Are the images shown representative of multiple assays for each tissue/target?

Response: for each tissue shown in Figure 7 at least four independent transgenic lines were analysed. The experiment was carried out twice with independent batches of plants and representative pictures are shown. This information has been added to the legend to Figure 7.

Would it be possible to redesign this figure to show representative images from TPS2, TPS8, and negative controls for each tissue type such that they were side by side instead of all grouped by gene and with no controls shown? I’m not sure that you can make claims comparing TPS2 and TPS8 expression based on the shown images alone.  The qPCR data is more convincing.  For instance, Figure 7J and 7T seem to show major differences in TPS2 vs TPS8 but the qPCR shows both genes high in roots. Could the apparent difference be due to camera setting (exposure time) or something? Is there a way to show how consistent these GUS experiments were from replicate to replicate?

Response: histological GUS stainings have only limited dynamic range making quantitative conclusions difficult. In general, a comparison between GUS staining and qPCR data is particularly difficult and differences in the outcome between the two approaches can be expected due to dynamics and resolution (e.g. bulk tissue vs. cell specificity). As we compare the expression patterns of TPS2 and TPS8, we feel that it does not add additional information to compare these to the wild type control as this does not display staining. The histological analysis was carried out on at least 4 independent transgenic tomato lines and the experiment was done twice. For sure, there is always some difference between the samples; however, Figure 7 represents the general qualitative picture consistent between lines and experiments. To our knowledge this is the way how these experiments are generally documented and we are not sure how it would be possible to incorporate replicates into the figure to further substantiate the findings.

Minor considerations:

The first half of the first paragraph could be omitted as knowledge of what a tomato is and what it is used for can be assumed.

Response: given the fact that IJMS is a multidisciplinary journal addressing a broad readership, we felt that giving a little broader introduction into the subject could make the study better accessible for non-experts in field. Therefore, we would prefer to keep this section.

Line 55: I suggest that you replace “will increase” with “are expected to increase”.

Response: done

Line 253: I believe this should be 5’ UTR (not 3’ UTR).

Response: right, corrected

Line 460: More detail is needed for the GUS methodology. How many replicates were imaged? What type of instrument and settings were used? What kind of controls?

Response: we have added additional experimental details to the method, including sample collection, buffer conditions and image acquisition.

Reviewer 2 Report

This paper is a correct presentation of interesting results of well-designed experiments. The Authors have characterized tomato genes involved in the production of trehalose. The dissaccharide is a well-known stress protectant, therefore the research topic is up-to-date and might attract the attention of the reseach community, especially in the era of global warning. They have found that only 2 (out of 10) TPS genes, but all TPP genes were able to rescue the corresponding yeast mutants. Furthermore, they have shown that the expression of the TPS genes are tissue specific and together with the TPP genes respond to temperature changes.
The experiments are well-planned, adequately described and the conclusions are logical and correct. This work may serve as a basic point to study the role of trehalose during stress and contribute to the development of more stress-resistant tomato.

Minor change: The graphs in Figure 3 seems to be mislabelled. I suppose the "Galactose control" is shown on the "b" graph not on "a" as indicated.

Author Response

Thanks to the reviewer for pointing this out. The error has been corrected in the revised version of the ms.